# Tumor-Derived Exosomal miR-143-3p Induces Macrophage M2 Polarization to Cause Radiation Resistance in Locally Advanced Esophageal Squamous Cell Carcinoma

**DOI:** 10.3390/ijms25116082

**Published:** 2024-05-31

**Authors:** Lin-Rui Gao, Jiajun Zhang, Ning Huang, Wei Deng, Wenjie Ni, Zefen Xiao, Mei Liu

**Affiliations:** 1Department of Radiation Oncology, National Cancer Center/National Clinical Research Center for Cancer/Cancer Hospital, Chinese Academy of Medical Sciences and Peking Union Medical College, Beijing 100021, China; gao_linrui@126.com (L.-R.G.); sherrydw@126.com (W.D.); niwenjie_1990@163.com (W.N.); 2Department of Hepatobiliary Surgery, National Cancer Center/National Clinical Research Center for Cancer/Cancer Hospital, Chinese Academy of Medical Sciences and Peking Union Medical College, Beijing 100021, China; zhangjiajun27886@163.com (J.Z.); 15093217398@163.com (N.H.); 3Laboratory of Cell and Molecular Biology, State Key Laboratory of Molecular Oncology, National Cancer Center/National Clinical Research Center for Cancer/Cancer Hospital, Chinese Academy of Medical Sciences and Peking Union Medical College, Beijing 100021, China

**Keywords:** radiotherapy, locally advanced esophageal squamous cell carcinoma, exosome, M2 macrophage polarization, miR-143-3p, miR-223-3p

## Abstract

We aimed to determine whether monitoring tumor-derived exosomal microRNAs (miRNAs) could be used to assess radiotherapeutic sensitivity in patients with locally advanced esophageal squamous cell carcinoma (ESCC). RNA sequencing was employed to conduct a comparative analysis of miRNA expression levels during radiotherapy, focusing on identifying miRNAs associated with progression. Electron microscopy confirmed the existence of exosomes, and co-cultivation assays and immunofluorescence validated their capacity to infiltrate macrophages. To determine the mechanism by which exosomal miR-143-3p regulates the interplay between ESCC cells and M2 macrophages, ESCC cell-derived exosomes were co-cultured with macrophages. Serum miR-143-3p and miR-223-3p were elevated during radiotherapy, suggesting resistance to radiation and an unfavorable prognosis for ESCC. Increased levels of both miRNAs independently predicted shorter progression-free survival (*p* = 0.015). We developed a diagnostic model for ESCC using serum microRNAs, resulting in an area under the curve of 0.751. Radiotherapy enhanced the release of miR-143-3p from ESCC cell-derived exosomes. Immune cell infiltration analysis at the Cancer Genome Atlas (TCGA) database revealed that ESCC cell-derived miR-143-3p triggered M2 macrophage polarization. Mechanistically, miR-143-3p upregulation affected chemokine activity and cytokine signaling pathways. Furthermore, ESCC cell exosomal miR-143-3p could be transferred to macrophages, thereby promoting their polarization. Serum miR-143-3p and miR-223-3p could represent diagnostic and prognostic markers for patients with ESCC undergoing radiotherapy. Unfavorable prognosis could be linked to the increased levels of ESCC cell-derived exosomal miR-143-3p, which might promote tumor progression by interacting with macrophages.

## 1. Introduction

Esophageal cancer is the ninth most common cancer, with a 5-year survival rate of only 20% globally [1]. Approximately half of all cases are reported in East and Central Asia, particularly in China, where esophageal squamous cell carcinoma (ESCC) is the predominant histological subtype [2]. At diagnosis, over 60% of patients present with locally advanced disease (T3, T4, or N^+^) or even more advanced stages [3,4].

Clinical guidelines recommend definitive radiotherapy as the standard of care for unresectable locally advanced ESCC [5,6,7]. However, we lack reliable diagnostic and prognostic biomarkers; therefore, the majority of patients with ESCC are diagnosed at an advanced disease stage. Despite the progress made in radiotherapy methods and treatment regimens, overall prognosis remains unfavorable [4,8,9]. The 5-year overall survival (OS) rate ranges from 30.1% to 45.6% [4,10,11,12,13]. Additionally, approximately 50% of patients with ESCC experience local-regional and distant failure following radiotherapy [14,15,16]. Therefore, it is imperative to determine the molecular mechanisms underlying ESCC progression and recurrence and identify dependable markers for diagnosis and prognosis.

Exosomes, tiny cell-derived vesicles measuring between 30 and 150 nm in diameter, play a vital role in modifying the tumor microenvironment (TME) and facilitating tumor dissemination through the transfer of signaling peptides, noncoding RNAs, or DNA to nearby cells or tissues [17,18]. MicroRNAs (miRNAs) are a group of short noncoding RNAs, ranging from 17 to 24 nt in size, which are major constituents of exosomes. These miRNAs are generated and released by a single cell, followed by internalization by neighboring or distant cells, exerting regulatory effects on the recipient cells [19,20]. MiRNAs modulate the expression of target mRNAs through binding to their 3′-untranslated regions and have been reported to contribute to the progression of different types of cancers [21]. 

The disturbance of exosomal miRNAs can profoundly influence the communication between cancer cells and the TME [22], ultimately enhancing the metastatic potential of cancer [23,24]. Furthermore, macrophages are the primary invasive immune-associated stromal cells present inside and around tumors [25,26]. In addition to regulating tumorigenesis, macrophages also influence distant metastasis and respond to immune checkpoint blockade therapies [27]. Various stimuli trigger the polarization of tumor-associated macrophages (TAMs), causing their differentiation into either M1 macrophages or M2 macrophages [25]. M2 macrophages promote progression and the formation of metastasis, as evidenced by their participation in remodeling the extracellular matrix, facilitating immune evasion, and fostering neovascularization [28,29].

However, the specific molecular mechanisms through which tumor-derived exosomal miRNAs interact with TAMs to induce radioresistance in locally advanced ESCC, and their subsequent impacts on tumor progression and metastasis, remain unexplored. In the present study, we examined the expression levels of miR-143-3p and miR-223-3p in patients with ESCC undergoing radiotherapy and investigated whether ESCC exosomes could promote the polarization of M2 macrophages by secreting miR-143-3p. Our objective was to identify a novel and specific biomarker, as well as to develop new predictive strategies to assess the risk of progression or metastasis in patients with ESCC after radiotherapy.

## 2. Results

### 2.1. Screening Phase

A preliminary investigation was carried out to explore the possible correlation between miRNAs and the sensitivity to radiation in individuals diagnosed with locally advanced ESCC. Three patients from the progression group and three patients from the progression-free group were selected for analysis. We used RNA-seq to compare the miRNA expression levels in serum samples at baseline and during radiotherapy. Appendix A shows the clinical features of these six individuals. The miRNAs satisfying the following criterion were included in the subsequent analysis: CPM (counts per million) ≥ 500 in all 12 serum samples. A total of 79 miRNAs satisfied the criterion. Based on their significant differences within and between groups (fold change ≥ 2, *p* < 0.05), four out of the seventy-nine miRNAs were recognized as potential candidate biomarkers for further validation (Figure 1A). Compared with those at baseline, the levels of miR-181a-5p (*p* = 0.016) and miR-223-3p (*p* = 0.012) were significantly decreased while miR-337-3p (*p* = 0.038) showed an increase in the progression-free group (Figure 1B–E). In the progression group, miR-143-3p (*p* = 0.038) was significantly upregulated (Figure 1F–I). These four candidate miRNAs were subsequently validated in a separate set of 80 serum samples obtained from 40 patients with ESCC.

### 2.2. The Levels of Serum miRNAs Are Associated with Progression-Free Survival of Patients with ESCC Receiving Radiotherapy

To further investigate the correlation between changes in miRNA expression during radiotherapy and recurrence/progression, we analyzed four specific miRNAs in 80 serum samples from 40 patients with ESCC. The median follow-up time of the entire group was 20 (6–74) months. To ensure consistency, the miRNA levels in all serum samples were standardized to that of miR-191-5p. The detection rate of miR-337-3p was found to be only 63.0% (34 out of 54 samples); therefore, it was excluded from subsequent analysis. Table 1 shows a summary of the patients’ clinical characteristics. Based on the duration of PFS, the patients were categorized into two groups: those with a progression-free period of more than 12 months and those who experienced recurrence/progression within 12 months. By the time of the final follow-up, a total of 19 patients had experienced recurrence/progression within the initial year after treatment. Based on the fold change in serum miRNA levels, the 40 patients could be categorized into two groups: a sensitive group (≤1.5-fold) and a resistant group (>1.5-fold). Univariate analysis did not reveal any correlation between the change in individual miRNA levels and recurrence (Table 1). Furthermore, the prediction accuracy of a single miRNA was found to be lower than 75%, with miR-143-3p at 65.0%, miR-181a-5p at 55.0%, and miR-223-3p at 72.5% (Figure 1J).

However, the alterations in the expression levels of miR-143-3p and miR-223-3p, which stratified the patients into a radiation-sensitive group (both ≤ 1.5-fold) and a resistant group (any > 1.5-fold or both > 1.5), significantly enhanced the prediction accuracy (82.5%; Figure 1K). Furthermore, our study demonstrated that the resistant group exhibited a significantly shorter OS (*p* = 0.024) and PFS (*p* = 0.008) when evaluated using the combination of miR-143-3p and miR-223-3p (Figure 1L,M). 

To further substantiate the potential role of miR-143-3p and miR-223-3p in predicting the recurrence/progression of ESCC, we conducted univariate (Table 1) and multivariate analyses (Appendix A) to evaluate the prognostic values of various commonly utilized clinicopathological characteristics. The multivariate analysis showed that the Karnofsky Performance Status (KPS), primary tumor volume, and the fold change in miR-143-3p and miR-223-3p were independently associated with OS and PFS. Furthermore, the ROC curve analysis of the KPS, primary tumor volume, and the fold change in miR-143-3p and miR-223-3p exhibited AUC values of 0.703 to predict OS and 0.751 to predict PFS, respectively (Figure 1N,O). 

### 2.3. Radiotherapy Promotes the Delivery of ESCC-Derived Exosomes Carrying miR-143-3p into the Tumor Microenvironment

To examine the possible influence of radiation on the expression of candidate miRNAs in ESCC cells, we measured miRNA expression levels in three different ESCC cell lines with and without radiation. The results indicated that radiation not only greatly enhanced the expression levels of miR-143-3p in all three ESCC cell lines but also promoted the secretion of exosomal miR-143-3p into the culture medium (CM) (Figure 2A–C). However, the impact of radiation on the expression levels of miR-181a-5p and miR-223-3p varied among the cell lines and the CM (Figure 2D–F). To determine whether miR-143-3p is a secretory miRNA, we examined the presence of miR-143-3p in the CM. Exosomes were isolated from the CM obtained from various ESCC cell lines using ultracentrifugation. Western blotting analysis confirmed the existence of the exosome markers HSP70, TSG101, and CD9 in the isolated exosomes. Additionally, the isolation of exosomes was further validated using a NanoSight device and TEM. Therefore, the hypothesis was proposed that miR-143-3p is secreted via exosomes derived from ESCC cells. Subsequently, ESCC cells were treated with GW4869, a drug that obstructs exosome biogenesis by inhibiting neutral sphingomyelinase 2 (nSMase2) and reduces the miRNA content in exosomes [30], resulting in the inhibition of exosome formation. GW4869 blocked exosome secretion (Figure 2G) and statistically significantly reduced the miR-143-3p levels in exosomes isolated from the CM (Figure 2J). After treating KYSE150 and KYSE510 cells with RNaseA (2 mg/mL), alone or in combination with Triton X-100 (0.1%), for 30 min, the levels of miR-143-3p in the CM were analyzed using qRT-PCR. RNase A cannot penetrate the membrane or extracellular vesicles (EVs) and can only digest the RNA outside of EVs. Following the application of RNase A, RNA was extracted from the supernatants to identify the presence of miR-143-3p. The disruption of the EV membrane with Triton X-100 and treatment with RNase A caused a notable reduction in the miR-143-3p level compared with that in the PBS group and the group treated with RNase A alone (Figure 2K), indicating that extracellular miR-143-3p was encased in a membrane and was not secreted directly.

### 2.4. MiR-143-3p Is Involved in the ESCC Immune Microenvironment

To further explore the molecular role of miR-143-3p in ESCC cells, cell lines (KYSE150, KYSE510, and KYSE180) stably transfected with pLV-miR-143-3p were established and their miR-143-3p expression levels were confirmed through qRT-PCR (Figure 3A). To compare the gene expression profiles between pLV-control and pLV-miR-143-3p KYSE150 cells, RNA sequencing analysis was performed. A volcano plot demonstrated that there were 468 downregulated genes and 1082 upregulated genes in pLV-miR-143-3p cells compared with those in the pLV-control cells (Figure 3B). The GSEA analysis revealed that miR-143-3p might play a role in the immune response by activating the humoral immune response and positively regulates interleukin 4 production and chemokine activity (Figure 3C–E). Furthermore, the functional enrichment analysis conducted in Metascape revealed that the pLV-miR-143-3p group exhibited associations with various biological processes, including the cell cycle, DNA replication, cytokine signaling in the immune system, and the inflammatory response (Figure 3F). These results suggested that miR-143-3p potentially plays a part in the regulation of the immune microenvironment in ESCC.

Notably, myeloid-derived macrophages, which possess innate immune functions, can undergo differentiation into TAMs, specifically M2 macrophages, within the TME. This phenotypic transformation of macrophages in the TME has been implicated in promoting tumor growth and progression [31,32]. Therefore, we conducted further bioinformatic analyses to determine the potential association between miR-143-3p and immune cell infiltration. Utilizing the TCGA cohort, cases of ESCC were divided into two groups: a group with high miR-143-3p expression and one with low miR-143-3p expression. The CIBERSORT-ABS algorithm was employed to examine the infiltration of intrinsic immune cells (including mast cells, macrophages, natural killer cells, eosinophils, and neutrophils) and adaptive immune cells (B cells and T cells). The results showed a significant correlation between the high expression of miR-143-3p and increased infiltration of M2 macrophages (Figure 3G). This correlation was further validated by the enrichment scores generated using the TIMER, QUANTISEQ, and CIBERSORT algorithms (Figure 3H–J).

Various molecular markers have been confirmed as biomarkers of M1 and M2 macrophages; therefore, we evaluated the expression correlation between the expression of these markers and miR-143-3p in the TCGA cohort. We observed that the high expression of miR-143-3p correlated negatively with the expression levels of M1 macrophage marker genes including *IL1B* (encoding interleukin 1 beta) and *iNOS* (encoding inducible nitrous oxide synthase) (Figure 3K,L) but was positively associated with the gene signatures of M2 macrophages including *TGFB1* (encoding transforming growth factor beta 1), *CD163* (encoding CD163 molecule), *CD301* (encoding C-type lectin domain containing 10A), *CD273* (encoding programmed cell death 1 ligand 2), *ALOX15* (encoding arachidonate 15-lipoxygenase), *CSF1R* (encoding colony stimulating factor 1 receptor), and *CCL18* (encoding C-C motif chemokine ligand 18) (Figure 3M–S). 

### 2.5. ESCC Cell-Derived Exosomal miR-143-3p Induces M2 Polarization of Macrophages

To explore the capability of exosomes derived from ESCC cells to promote the M2 polarization of macrophages, we utilized the human THP-1 cell line as a representative mononuclear macrophage line. After incubation with PMA, the THP-1 cells differentiated into M0 macrophages, which were identified by their adherent appearance. Thereafter, we assessed the impact of exosomes derived from ESCC cells on the polarization of macrophages towards the M2 phenotype. In a co-cultivation assay, ESCC cell-derived exosomes labeled with EvLINK 505 (green) were internalized by CellINK 555 labeled-macrophages (red) (Figure 4A). Next, transcriptome sequencing was performed after extracting total RNA from macrophages that were co-cultivated with exosomes derived from pLV-control or pLV-miR-143-3p KYSE150 cells. Differential expression analysis based on read counts identified 131 upregulated genes and 81 downregulated genes in the miR-143-3p-overexpressing group (Figure 4B). The GSEA analysis demonstrated that when macrophages were incubated with exosomes derived from pLV-miR-143-3p KYSE150 cells, they exhibited enrichment in epithelial mesenchymal transition and the TGF-β signaling pathway, which are considered characteristics of M2 macrophage differentiation (Figure 4C,D). According to Metascape enrichment analysis, the co-cultivation of these macrophages with ESCC cell-derived exosomal miR-143-3p resulted in the enrichment of the inflammatory response, extracellular matrix organization, and regulation of the immune system (Figure 4E). In addition, we observed an increase in the expression of M2 markers, including arginase-1 (Arg-1), mannose receptor C-type 1 (MRC1, also known as CD206), and IL10, in macrophages that were co-cultured with exosomes obtained from pLV-miR-143-3p cells, in contrast to macrophages treated with exosomes from pLV-control cells or cells incubated with PBS (Figure 4F–H). Conversely, the expression levels of M1 markers, including iNOS, CD80, and IL-1β, were significantly decreased in macrophages that were co-cultured with exosomes obtained from pLV miR-143-3p cells (Figure 4I–K).

## 3. Materials and Methods

### 3.1. Clinical Specimens and Ethical Approval

Serum samples were collected from patients with ESCC who underwent definitive radiotherapy at the Cancer Hospital, CAMS between January 2015 and December 2019 (NCT05543057). The clinicopathological diagnoses were verified by at least two pathologists following the American Joint Committee on Cancer (AJCC) guidelines. A total radiotherapy dose of 50.0–60.62 Gy (equivalent dose in 2 Gy fractions (EQD2); median dose: 60.62 Gy) was delivered once a day for 5 days each week. Among the patients, 21 patients received concurrent chemotherapy.

Serum samples were collected at the following time points: pre-radiotherapy (baseline) and during radiotherapy (20–23 fractions). Serum was extracted from blood samples through centrifugation at 3000× *g* for 10 min. The Independent Ethics Committee of CAMS approved the project (No. 22/036-3237). All patients provided informed consent before enrollment in this study.

### 3.2. Follow-Up

Patients underwent weekly assessments during the treatment period, every 3–6 months in the initial 2 years post-treatment, every 6–12 months for the following 3 years, and annually thereafter. The assessment included a comprehensive evaluation of symptoms, such as cough, fever, hoarseness, dysphagia, and chest tightness, as well as a thorough examination of each patient’s medical history. A variety of diagnostic tests were performed, encompassing blood examinations (such as complete blood count and a basic metabolic panel); analysis of tumor markers; contrast-enhanced computed tomography (CT) scans covering the neck, thorax, and abdomen; ultrasound imaging of the neck and abdomen; upper gastrointestinal contrast studies; bone scans (if there were indications such as bone pain or abnormally elevated alkaline phosphatase levels); and CT or magnetic resonance imaging (MRI) scans of the brain (if symptoms related to the central nervous system were observed). Additionally, endoscopy, endoscopic ultrasound, Fludeoxyglucose F18-positron emission tomography (18F-FDG PET(-CT)), scans, and fine-needle aspiration cytology were conducted if necessary.

### 3.3. Cell Culturing, RNA Isolation, and Quantitative Real-Time Reverse-Transcription PCR (qRT-PCR)

The human ESCC cell lines KYSE150, KYSE510, and KYSE180 and the human monocytic cell line THP-1 were cultivated at 37 °C in Roswell Park Memorial Institute (RPMI) 1640 medium supplemented with 10% fetal bovine serum (FBS), 100 U/mL streptomycin, and 100 U/mL penicillin. ESCC cells were subjected to radiation (8 Gy using 6 MV X-rays), subsequently incubated at 37 °C for 72 h, and then collected. To stimulate their transformation into macrophages, THP-1 (1 × 10^6^) cells were exposed to 100 ng/mL of phorbol 12-myristate 13-acetate (PMA; Abcam, Cambridge, UK) for 72 h.

The TRIzol Reagent (Invitrogen, Carlsbad, CA, USA) was employed to extract total RNA from cultured cells, which was then reverse-transcribed to cDNA using M-MLV Reverse Transcriptase (Promega, Madison, WI, USA). The StepOne Plus Real-Time PCR System (Applied Biosystems, Foster City, CA, USA) was used to conduct quantitative real-time PCR, following the manufacturer’s protocol, using Power SYBR Green PCR Master Mix (Applied Biosystems). Fold changes in gene expression were determined using the 2^−ΔΔCt^ method, with triplicate analysis performed for each sample [33]. *ACTB* (encoding β-actin) was employed as the internal control for normalization of the qRT-PCR data.

### 3.4. miRNA-Specific qRT-PCR

The TRIzol LS Reagent (Invitrogen) was employed to extract total RNA from serum, which was then converted into cDNA using an miScript II RT Kit (Qiagen, Hilden, Germany). A NanoDrop ND-2000 spectrophotometer (Thermo Scientific, Wilmington, DE, USA) was used to determine the RNA concentration and quality. The StepOne Plus Real-Time PCR System (Applied Biosystems) was used to conduct qPCR, following the manufacturer’s protocol and utilizing the miScript SYBR Green PCR Kit (Qiagen). Fold changes in gene expression were determined using the 2^−ΔΔCt^ method, with triplicate analysis performed for each sample [33]. The primers used in this study are listed in Appendix A. MiR-191-5p served as the internal control for normalization in both serum and culture medium [34] while U6 was employed as the internal control for normalization in cells.

### 3.5. RNA Sequencing

The TRIzol LS Reagent was employed to extract total RNA from serum. The BGISEQ-500 sequencing platform (Beijing Genomics Institute (BGI), Wuhan, China) was used to sequence serum miRNAs. We measured serum miRNAs from three patients from the progression group and three patients from the progression-free group at different time points: pre-radiotherapy (baseline) and during radiotherapy (20–23 fractions). Standardization of miRNA expression levels was carried out using counts per million. The DESeq2 R package was used to perform differential expression analysis of the two groups. For subsequent analysis, the miRNAs had to have a minimum of 500 expression counts in all samples (12/12). MiRNAs with a cut-off value of |log2 fold change (FC)| ≥ 1.0 and *p* < 0.05 were considered as differentially expressed miRNAs.

Samples from KYSE150 and THP-1 cells were prepared, and the TRIzol reagent was used to extract the total RNA. Raw sequencing data obtained from the BGISEQ-500 sequencing system underwent filtration to remove reads with low quality and high levels of unknown bases. The DESeq2 R software package (version 1.44.0) was employed to detect differentially expressed genes using a threshold of |log2 FC| ≥ 1.0 and *p* < 0.05.

### 3.6. Western Blotting

Cells were collected and lysed in Radioimmunoprecipitation assay (RIPA) buffer (Sigma, St Louis, MO, USA), followed by electrophoretic separation and transfer onto a membrane. Western blotting analysis was performed following established procedures as described previously [35]. The primary antibodies used recognized heat shock protein 70 (HSP70) (4873; 1:1000; Cell Signaling Technology, Danvers, MA, USA), tumor susceptibility 101 (TSG101) (14497; 1:1000; Proteintech, Rosemont, IL, USA), and CD9 (92726; 1:1000; Abcam, Cambridge, MA, USA).

### 3.7. Isolation and Analysis of Exosomes

A total of 6 × 10^6^ cells were cultured in medium containing 10% exosome-depleted FBS. After 72 h, 30 mL of supernatant was obtained and subjected to centrifugation at 3500× *g* for 30 min. This resulting supernatant was then filtered through 0.22 μm filters and subjected to ultracentrifugation at 120,000× *g* for 90 min. Exosomes were quantified using a NanoSight NS300 instrument (Malvern Instruments Ltd., Malvern, UK) equipped with NTA 3.0 analytical software (Malvern Instruments Ltd.).

Exosomes were wicked off to create a thin layer before addition of a thin layer of 2% phosphotungstic acid on the copper grid. Grids were allowed to dry overnight at room temperature, and transmission electron microscopy (TEM, HT7800, HITACHI, Tokyo, Japan) was performed the next day. 

### 3.8. Retroviral Infection

ESCC cells were transfected with either an empty lentiviral vector (pLV-control) or a lentiviral vector overexpressing pre-miR-143-3p (pLV-miR-143-3p). GeneChem Company (Shanghai, China) successfully performed the plasmid construction and lentivirus packaging, which were subsequently utilized for cell infection following the guidelines provided by the manufacturer. At 48 h post-infection, cells were selected using 1 μg/mL puromycin in the culture medium, with selection continuing for 14 days. Then, stable transfectants were harvested for further experimentation.

### 3.9. Immunofluorescence (IF)

The CellINK 555 cell-labeling kit (CellINK, Gothenburg, Sweden) and the EvLINK 505 exosome-labeling kit (TINGO Exosomes Technology, Tianjin, China) were employed for cell and exosome labeling according to the manufacturer’s protocol using the following antibodies: CellINK 555 at a dilution of 1:100 and EvLINK 505 at a dilution of 1:100. Afterwards, the cells/exosomes were counterstained using 4′,6-diamidino-2-phenylindole (DAPI) (1:1000, Invitrogen). This was used to counterstain the cells/exosomes at room temperature for 10 min. Images were acquired using a confocal microscope (Leica, Wetzlar, Germany).

### 3.10. Statistical Analysis

The endpoints in this study included OS and progression-free survival (PFS). The OS was defined as the duration from diagnosis to death from any cause or censoring at the last follow-up. We defined PFS as the duration from diagnosis to the onset of disease progression, relapse, or death from any cause.

Crude survival was calculated using the Kaplan–Meier method and compared using the log-rank test. The miRNA signature was assessed for its independent prognostic value using multivariate Cox proportional hazards regression analysis. Feasibility was determined using a receiver operating characteristic (ROC) curve and the derived area under the ROC curve (AUC).

To obtain the mRNA and miRNA expression data for esophageal cancer, we utilized the Cancer Genome Atlas (TCGA) database (https://cancergenome.nih.gov/tcga/, accessed on 24 June 2023). To explore the enrichment of particular sets of genes, Gene Set Enrichment Analysis (GSEA) software (version 3.0) was employed (http://software.broadinstitute.org/gsea/index.jsp, accessed on 25 June 2023). Metascape was employed to perform enrichment analyses for differentially expressed genes (http://metascape.org/, accessed on 30 June 2023). The infiltration levels of immune cells were predicted using TCGA data together with CIBERSORT-ABS, TIMER, QUANTISEQ, and CIBERSORT algorithms. All the R packages and analysis methods used were executed in R version 4.0.3 (https://www.r-project.org/, accessed on 30 June 2023), and *p* < 0.05 was considered statistically significant.

## 4. Discussion

In the current study, we demonstrated that elevated levels of serum miR-143-3p and miR-223-3p in patients with ESCC during radiotherapy are indicative of radioresistance and subsequent progression. The developed radiotherapy sensitivity-stratified prediction model might enable personalized therapy for patients with ESCC. Furthermore, we revealed that ESCC-derived exosomal miR-143-3p could be internalized by macrophages, thereby inducing polarization towards the M2 phenotype, which could potentially facilitate tumor progression and recurrence. These results indicated the potential significance of ESCC-derived exosomal miR-143-3p as a non-invasive biomarker to predict responses to radiotherapy and to determine patient outcomes.

As reported, approximately 50% of patients with ESCC experience progression or recurrence after definitive radiotherapy [14,15,16]. Given the relatively high risk of tumor recurrence in these patients, it might be advantageous to explore more intensive treatment options such as high-dose radiotherapy, consolidation therapy, or salvage therapy. Consequently, the development of effective predictive methods has the potential to offer risk-stratified and response-adapted treatment strategies for patients. Nonetheless, accurately evaluating the response to definitive therapy can be challenging [36], mainly because the majority of patients with inoperable lesions are unable undergo complete surgical excision after radiotherapy [37]. Additionally, previous studies indicated that endoscopic biopsy and current imaging techniques, such as CT, 18F-FDG PET(-CT), and endoscopic ultrasound (EUS), when employed separately, might not be dependably to detect complete responders [38,39,40,41,42,43]. Hence, there is a need for more accurate diagnostic tests to enhance restaging precision and identify patients who are unresponsive to treatment. Liquid biopsies offer a less invasive approach and provide a method to obtain tumor-related data from bodily fluids, which can be used to direct molecular-focused treatment strategies for patients who present difficulties for biopsy [44]. 

Numerous liquid biopsies, such as blood-based cell-free DNA (cfDNA), EV, and DNA methylation analyses, have been investigated as cancer screening techniques [44]. These methodologies hold promise in accurately identifying cancer recurrence, thereby enabling timely intervention. In our study, we assessed the alterations in serum miRNA levels during radiotherapy. It has been reported that the levels of miRNAs in cells and those derived from serum exosomes are essentially concordant [45]. The upregulation of serum miR-143-3p and miR-223-3p levels during radiotherapy could serve as an early indicator of radiosensitivity in ESCC, displaying an AUC ≥ 0.7 in predicting both OS and PFS. Moreover, miRNA detection in this study was carried out during radiotherapy (approximately two-thirds of definitive radiotherapy), which offered the possibility of reducing the interval between consolidation or salvage therapy, thereby mitigating the risk of therapeutic complications and mortality.

Accumulating evidence demonstrates that tumor-derived exosomes commonly transmit miRNAs to recipient cells, resulting in the suppression of their target genes [46,47]. However, limited research has been conducted on the influence of ESCC-derived exosomal miRNAs on macrophage polarization in the recurrence and progression of ESCC. Notably, serum samples from patients who experienced progression exhibited higher levels of exosomal miR-143-3p during radiotherapy than those at baseline, suggesting its potential utility in the development of liquid biopsy techniques. A study also reported a notable increase of hsa-miR-143-3p in ESCC tissues compared with that in adjacent normal tissue, along with its association with prognosis [48], which supported our results. Moreover, given the multitude of genomic alterations and the restricted effectiveness of therapies targeting cancer cells, recent research efforts have concentrated on acquiring a broader comprehension of the TME and potential strategies to alter it to facilitate cancer treatment [49]. The TME also contains exosomes, which are known to carry components that modulate various tumor processes such as metastasis, angiogenesis, and drug resistance [50].

Within the TME, exosomes have a vital function in facilitating the transfer of functional biomolecules, including miRNAs, to recipient cells. Given their predominance in the TME [51], TAMs have consistently been associated with the regulatory interaction between tumor cells and the TME. This interaction is facilitated by exosomal miRNAs, ultimately promoting the development and progression of tumors [23,52]. Moreover, increasing evidence suggests that elevated levels of infiltrating immune cells, particularly macrophages, are associated with the therapeutic response and prognosis in patients with ESCC [27]. In the present study, we demonstrated that miR-143-3p encapsulated within exosomes derived from ESCC cells can be transferred to macrophages. Subsequently, the ESCC cell-derived exosomal miR-143-3p induces the polarization of M2 macrophages. The polarization of TAMs toward the M2 phenotype is crucial for the regulation of tumor growth, migration, and angiogenesis. This M2 polarization stimulates macrophages to produce growth factors and cytokines, contributing to the intricate modulation of tumor-related processes [21,32]. This aligns with the report that exosomal miRNAs can be effectively transported into target cells to control their biological processes [46]. 

This study had some limitations, including its single-center design, a relatively small sample size, and the lack of a validation dataset. A study with a multi-center large sample size (e.g., n > 100) with longer follow-ups is needed to strengthen the generalizability of the findings. In addition, the primary constituents isolated through the methodologies outlined in our study are referred to as highly-enriched in exosomes because there is a possibility of contaminating other EVs. Most importantly, a more thorough exploration of the mechanisms by which ESCC cell-derived exosomal miR-143-3p induces M2 polarization in macrophages, along with its subsequent effects on protein expression and associated pathways, is necessary.

## 5. Conclusions

Our results emphasize the possible importance of serum miR-143-3p and miR-223-3p in diagnosing and predicting the outcomes of patients with ESCC receiving radiotherapy, providing valuable insights for future treatment strategies. Additionally, upregulated ESCC cell-derived exosomal miR-143-3p might induce M2 macrophage polarization, thereby facilitating tumor progression.

## Figures and Tables

**Figure 1 ijms-25-06082-f001:**
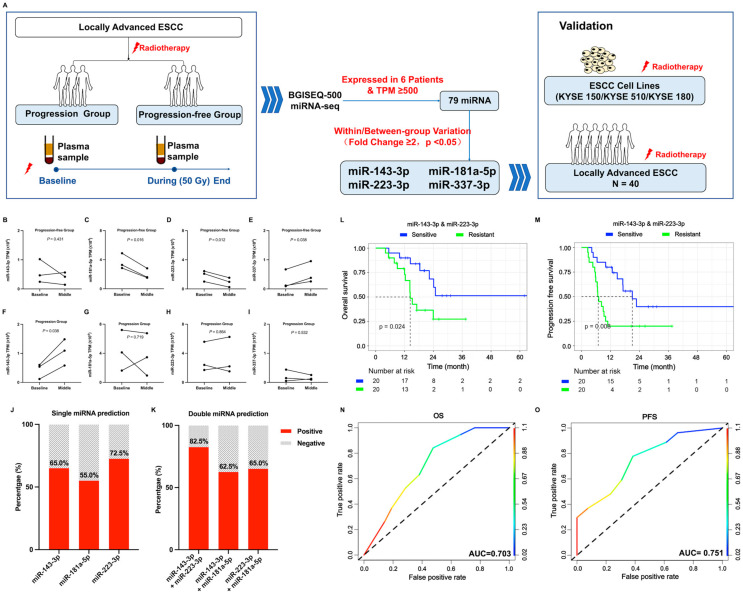
The expression of serum miRNAs is associated with the survival of patients with ESCC treated with radiotherapy. (**A**) Schematic diagram of the study workflow. (**B**–**I**) The relative expression levels of miR-143-3p, miR-181a-5p, miR-223-3p, and miR-337-3p at baseline and during radiotherapy in the progression-free group (**B**–**E**) and the progression group (**F**–**I**), respectively. (**J**,**K**) The prediction accuracy for radiotherapy based on the fold change in a single miRNA (**J**) or double miRNAs (**K**). (**L**,**M**) The overall survival (OS, **L**) and progression-free survival (PFS, **M**) of patients in the radiation-sensitive group (both ≤ 1.5-fold) and resistant group (any > 1.5-fold) predicted using miR-143-3p and miR-223-3p. (**N**,**O**) The ROC curve of KPS, primary lesion volume, and the fold change in miR-143-3p and miR-223-3p in predicting OS (**N**) and PFS (**O**). Abbreviations: miRNA, microRNA; ESCC, esophageal squamous cell carcinoma; ROC, receiver operating characteristic; KPS, Karnofsky Performance Status; BGISEQ-500, Beijing Genomics Institute high-throughput sequencing; miRNA-Seq, microRNA sequencing; TPM, Transcripts Per Kilobase Million.

**Figure 2 ijms-25-06082-f002:**
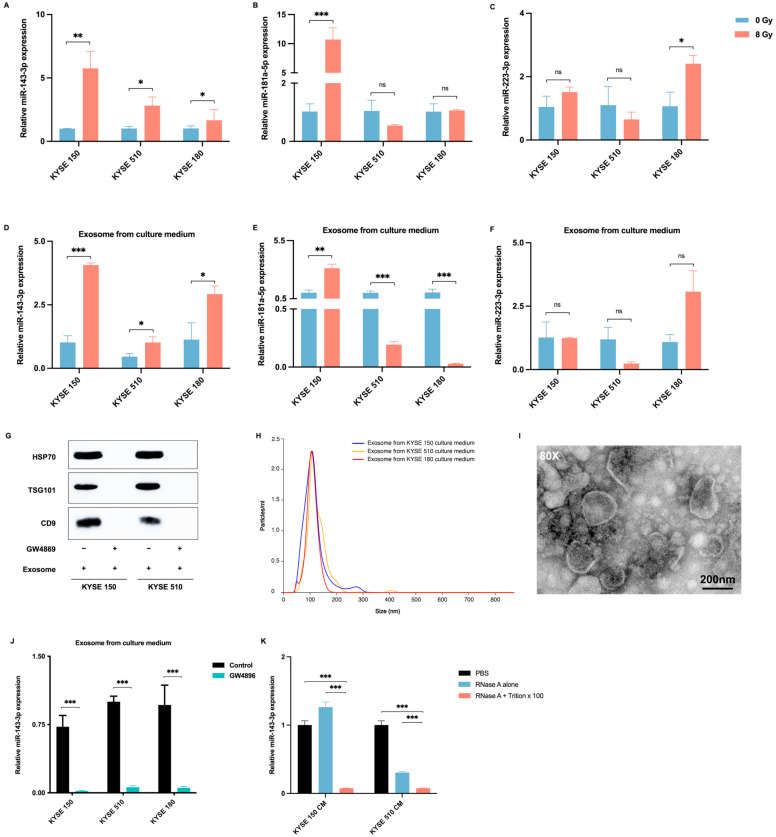
Radiotherapy promotes ESCC-derived exosomal miR-143-3p secretion. (**A**–**F**) qRT-PCR analysis of the miR-143-3p, miR-181a-5p, and miR-223-3p levels in ESCC cells (KYSE150, KYSE510, and KYSE180) cell lines (**A**–**C**) or exosomes from culture medium (**D**–**F**) treated with or without 8 Gy of radiation and allowed to recover for 72 h; *n* = 3 samples per group. (**G**) Western blotting analysis to detect typical exosomal biomarkers (HSP70, TSG101, and CD9) in exosomes. KYSE150 and KYSE 510 cells were treated with 10 μM GW4869 (an inhibitor of exosome secretion) or dimethyl sulfoxide (DMSO) (as a control). After 72 h, the culture medium was collected. Exosomes were isolated through ultracentrifugation. (**H**) NanoSight particle tracking analysis of the exosome size distribution and numbers derived from the culture medium of KYSE150, KYSE510, and KYSE180 cells. (**I**) Phenotypic analysis of exosomes derived from the culture medium of KYSE150 cells using electron microscopy. (**J**) qRT-PCR analysis of miR-143-3p expression in the culture medium of KYSE150, KYSE510, and KYSE180 cells depleted of exosomes using GW4869 or DMSO (control). (**K**) qRT-PCR analysis of the miR-143-3p levels in the culture medium of KYSE150 and KYSE510 cells treated with RNaseA (2 mg/mL) alone or in combination with Triton X-100 (0.1%) for 30 min. Abbreviations: ESCC, esophageal squamous cell carcinoma; qRT-PCR, quantitative real-time reverse-transcription PCR; PBS, phosphate-buffered saline; ns, not significant. (***: *p* < 0.001, **: *p* < 0.01, *: *p* < 0.05, ns: not significant).

**Figure 3 ijms-25-06082-f003:**
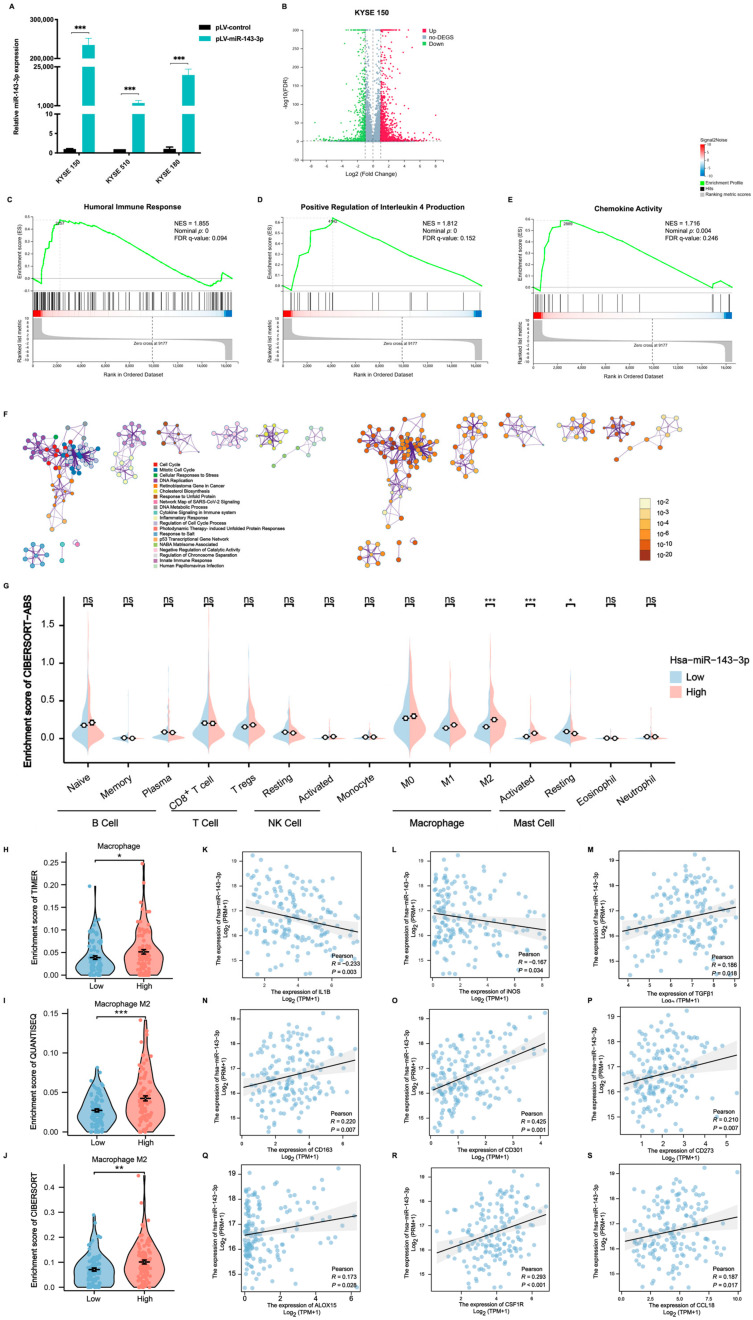
MiR-143-3p is involved in the ESCC immune microenvironment. (**A**) qRT-PCR analysis of the miR-143-3p levels in pLV-miR-143-3p- and pLV-control-transfected KYSE150, KYSE510, and KYSE180 cells. (**B**) Volcano plot showing the differential genes between the pLV-miR-143-3p and pLV-control-transfected KYSE150 cells. (**C**–**E**) Gene set enrichment analysis (GSEA) revealing the activated immune response induced by activation of the humoral immune response (**C**), positive regulation of interleukin 4 production (**D**), and chemokine activity (**E**) in pLV-miR-143-3p-tranfected KYSE150 cells. (**F**) The Metascape enrichment analysis results for differentially expressed genes across pLV-miR-143-3p and pLV-control-transfected KYSE150 cells. (**G**) Boxplot demonstrating the infiltration levels of immune cells in the high-miR-143-3p and low-miR-143-3p groups using the CIBERSORT-ABS algorithm. (**H**–**J**) Boxplot exhibiting the infiltration levels of macrophages in the high-miR-143-3p and low-miR-143-3p groups using TIMER (**H**), QUANTISEQ (**I**), and CIBERSORT (**J**) algorithms. (**K**–**M**) Pearson correlation analysis of expression levels between miR-143-3p and M1 macrophage markers (*IL1B* and *iNOS*) based on TCGA data. (**N**–**S**) Pearson correlation analyses of the expression levels between miR-143-3p and M2 macrophage markers (*TGFB1*, *CD163*, *CD301*, *CD273*, *ALOX15*, *CSF1R*, and *CCL18*) based on TCGA data. Abbreviations: ESCC, esophageal squamous cell carcinoma; qRT-PCR, quantitative real-time reverse-transcription PCR; TGCA, The Cancer Genome Atlas; ns, not significant; TPM, Transcripts Per Kilobase Million. (***: *p* < 0.001, **: *p* < 0.01, *: *p* < 0.05, ns: not significant).

**Figure 4 ijms-25-06082-f004:**
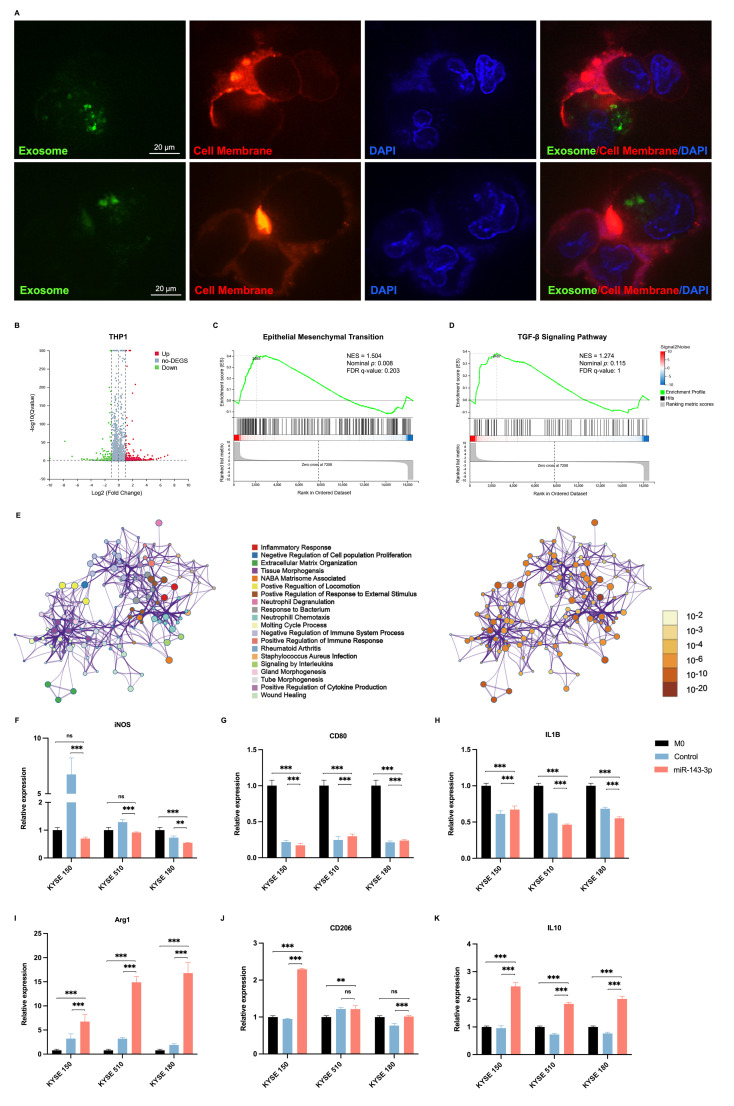
ESCC cell-derived exosomal miR-143-3p induces M2 polarization of macrophages. (**A**) Two representative immunofluorescence images revealing the internalization of EvLINK 505-labeled KYSE150 cell-derived exosomes (green) by PMA-treated THP-1 cells. (**B**) Volcano plot showing the differentially expressed genes in THP-1 cells treated with exosomes derived from pLV-miR-143-3p and pLV-control-transfected KYSE150 cells. (**C**,**D**) Gene set enrichment analysis (GSEA) revealing activation of epithelial mesenchymal transition (**C**) and the TGFβ signaling pathway (**D**) in THP-1 cells treated with exosomes derived from pLV-miR-143-3p-transfected KYSE150 cells. (**E**) The Metascape enrichment analysis for differentially expressed genes in THP-1 cells treated with exosomes derived from pLV-miR-143-3p and pLV-control-transfected KYSE150 cells. (**F**–**H**) qRT-PCR analysis of M1 macrophage markers (*iNOS*, *CD80*, and *IL1B*) expression levels in THP-1 cells treated with exosomes derived from pLV-miR-143-3p and pLV-control-transfected KYSE150, KYSE510, and KYSE180 cells. (**I**–**K**) qRT-PCR analysis of M2 macrophage marker (*ARG1*, *CD206*, and *IL10*) expression levels in THP-1 cells treated with exosomes derived from pLV-miR-143-3p and pLV-control-transfected KYSE150, KYSE510, and KYSE180 cells. Abbreviations: ESCC, esophageal squamous cell carcinoma; PMA, phorbol 12-myristate 13-acetate; qRT-PCR, quantitative real-time reverse-transcription PCR; UP, upregulated; DEG, differentially expressed gene; DOWN, downregulated; ns, not significant. (***: *p* < 0.001, **: *p* < 0.01, ns: not significant).

**Table 1 ijms-25-06082-t001:** Cohort characteristics of 40 patients with locally advanced esophageal squamous cell carcinoma treated with radiotherapy.

	N	%	OS	PFS
			*p*	*p*
Age (year)			0.108	0.040
≥65	18	45.0		
<65	22	55.9		
KPS			0.019	0.014
≥90	17	42.5		
<90	23	57.5		
Stage (AJCC 6th)			0.064	0.035
I–II	8	20.0		
III–IV	32	80.0		
Location			0.392	0.128
Cervical and upper	8	20.0		
Middle and lower	32	80.0		
Length			0.016	0.001
≤5 cm	18	45.0		
>5 cm	22	55.0		
Primary tumor volume			0.019	0.007
≤50 cm^3^	18	45.0		
>50 cm^3^	22	55.0		
Therapeutic approach			0.838	0.610
RT	19	47.5		
CRT (Single)	7	17.5		
CRT (Doublet)	14	35.0		
Response			0.566	0.758
ORR	20	50.0		
SD	14	35.0		
PD	1	2.5		
NA	5	12.5		
Fold change in miR-143-3p			0.872	0.443
≤1.5	27	67.5		
>1.5	13	32.5		
Fold change in miR-181a-5p			0.382	0.833
≤1.5	19	47.5		
>1.5	21	52.5		
Fold change in miR-223-3p			0.069	0.128
≤1.5	24	60.0		
>1.5	16	40.0		
Fold change in miR-143-3p and miR-223-3p			0.024	0.008
Both ≤ 1.5	20	50.0		
Any one > 1.5 and both > 1.5	20	50.0		
Fold change in miR-143-3p and miR-181a-5p			0.318	0.456
Both ≤ 1.5	18	45.0		
Any one > 1.5 and both > 1.5	22	55.0		
Fold change in miR-181a-5p and miR-223-3p			0.034	0.401
Both ≤ 1.5	15	37.5		
Any one > 1.5 and both > 1.5	25	62.5		

## Data Availability

The corresponding author has full access to all the data in the study and final responsibility for the decision to submit for publication. Further inquiries can be directed to the corresponding author.

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
