# Peer review of "Tumor-Derived Exosomal miR-143-3p Induces Macrophage M2 Polarization to Cause Radiation Resistance in Locally Advanced Esophageal Squamous Cell Carcinoma"

_ijms, 2024, doi:10.3390/ijms25116082_

Round 1

Reviewer 1 Report

Comments and Suggestions for Authors

This is a well-written and thorough paper, with interesting results. If the few concerns listed below are sufficiently addressed, this paper would merit publication in this journal. 

1.  There does not appear to be any loading controls for the Western blots, in either Fig 2D or in the file of "Raw Data of Western Blots Images." Loading controls need to be included to ensure proper loading of samples, transfer, etc. These should be added to this manuscript.

2.   It is unclear why the exosomes from serum samples were not characterized as rigorously as those from the cell lines. This lowers the confidence in what was isolated from the serum samples were, indeed, exosomes and not microvesicles (MVs), or a combination of those extracellular vesicles (EVs). While the NTA can quantify and characterize EVs, that is done only by size. Given the wide overlap in size ranges of exosomes (~30-150 nm) and MVs (~100-1000nm), one cannot be sure there is no "contamination" of one of those EVs with the other. Moreover, the EvLINK 505 exosome-labeling kit is questionable for labeling only exosomes. The spec sheet claims it couples the fluorescent probe to the outer membrane surface of exosomes, however there are no details of how the coupling happens and what factors (e.g. lipid bilayer proteins, etc) are involved. Given the great similarities between membranes of all EVs, one cannot be sure this kit is isolating exosomes and not other EVs as well. While this uncertainty (exosome vs multiple EVs) does not diminish the findings, it does mean the language here should be changed from stating only exosomes were tested, to stating that EVs (which would include exosomes and MVs) were tested. 

3. Section 3.4 includes a good deal of speculation of effects of miR-143-3. Why not actually test some of these effects in the cells - i.e. block/inhibit miRNA-143 to see if alters levels of IL-4 and/or other chemokines? This would strengthen the various claims made here, that are not actually proven with the data currently presented. 

Author Response

Dear Editor and reviewers,

We sincerely appreciate the careful reviews of our work and the reviewers’ thoughtful comments. We have responded to the reviewers’ comments and have updated the manuscript accordingly. We highlighted our answers in blue for easily reading in Response letter.

However, considering that there is only a 10-day time for revisions, there hasn't been sufficient time to conduct additional experiments. We have made our best effort to respond to the reviewers' comments. If further modifications and improvements are necessary, please do not hesitate to contact us with any further questions.

Furthermore, this manuscript was copyedited by a professional English editing service (Elixigen Co.) that specializes in scientific papers before submission. We have conducted a thorough review and further revised the language. The revised manuscript has been sent to Elixigen Co. for proofreading.

We continue to appreciate your consideration of our manuscript for publication in your esteemed journal.

Sincerely,

Mei Liu, PhD

Laboratory of Cell and Molecular Biology

State Key Laboratory of Molecular Oncology

National Cancer Center/National Clinical Research Center for Cancer/Cancer Hospital

Chinese Academy of Medical Sciences & Peking Union Medical College

Beijing 100021

P.R. China

Comments and Suggestions for Authors

This is a well-written and thorough paper, with interesting results. If the few concerns listed below are sufficiently addressed, this paper would merit publication in this journal. 

  1. There does not appear to be any loading controls for the Western blots, in either Fig 2D or in the file of "Raw Data of Western Blots Images." Loading controls need to be included to ensure proper loading of samples, transfer, etc. These should be added to this manuscript.

Response: Thank you for your questions. HSP70, TSG101, and CD9 have been consistently identified as enriched exosome marker proteins for many years [1-4]. The current study is centered on qualitative detection of exosomes through analysis, hence the omission of loading controls in the assessment. Previous research has suggested diminished expression levels of Actin, GAPDH, or Tubulin in exosomes relative to standard cell samples, leading to the exclusion of loading control detection in series of studies [4-8].

  1.  It is unclear why the exosomes from serum samples were not characterized as rigorously as those from the cell lines. This lowers the confidence in what was isolated from the serum samples were, indeed, exosomes and not microvesicles (MVs), or a combination of those extracellular vesicles (EVs). While the NTA can quantify and characterize EVs, that is done only by size. Given the wide overlap in size ranges of exosomes (~30-150 nm) and MVs (~100-1000nm), one cannot be sure there is no "contamination" of one of those EVs with the other. Moreover, the EvLINK 505 exosome-labeling kit is questionable for labeling only exosomes. The spec sheet claims it couples the fluorescent probe to the outer membrane surface of exosomes, however there are no details of how the coupling happens and what factors (e.g. lipid bilayer proteins, etc) are involved. Given the great similarities between membranes of all EVs, one cannot be sure this kit is isolating exosomes and not other EVs as well. While this uncertainty (exosome vs multiple EVs) does not diminish the findings, it does mean the language here should be changed from stating only exosomes were tested, to stating that EVs (which would include exosomes and MVs) were tested. 

Response: Thanks for your comments. We do agree that relying solely on NanoSight or the EvLINK 505 exosome-labeling kit for analysis may not conclusively establish the nature of particles as exosomes or EVs. Nevertheless, particles acquired via the same extraction technique have been authenticated as exosomes via electron microscopy and detection of exosome marker proteins. Western blot, NanoSight, and transmission electron microscopy are conventional and standard methods for identifying exosomes [7, 9-11]. The primary constituents isolated through the methodologies outlined in our study are exosomes. Althoug there is a possibility of other EVs contamination, the EVs labeled by the EvLINK 505 exosome-labeling kit here were mainly exosomes.

  1. Section 3.4 includes a good deal of speculation of effects of miR-143-3. Why not actually test some of these effects in the cells - i.e. block/inhibit miRNA-143 to see if alters levels of IL-4 and/or other chemokines? This would strengthen the various claims made here, that are not actually proven with the data currently presented. 

Response: We do agree with your comments. Our research primarily centers on investigating the impact of miR-143-3p on macrophages. Through RNA sequencing analysis, it has been indicated that miR-143-3p may modulate the immune microenvironment of ESCC by potentially enhancing IL4 productionand chemokine activity. Previous studies have also suggested a potential association between miR-143 and IL-4 [12, 13] or chemokine-mediated signaling pathways [14-16], albeit within non-oncological contexts.

Due to the constrained timeframe of 10 days for revisions, it may present difficulties in obtaining the required materials and carrying out experiments to block or inhibit miRNA-143. We added the limitations in the revised manuscript as follows (Lines 497-494):

Furthermore, a more thorough exploration of the mechanisms by which ESCC cell-derived exosomal miR-143-3p induces M2 polarization in macrophages, along with its subsequent effects on protein expression and associated pathways, is necessary.

We sincerely appreciate your professional comments and patient help. Should you still recommend further exploration of this aspect of the study, please do let us know promptly.

Reference

  1. Escola, J.M., et al., Selective enrichment of tetraspan proteins on the internal vesicles of multivesicular endosomes and on exosomes secreted by human B-lymphocytes. J Biol Chem, 1998. 273(32): p. 20121-7.
  2. Thery, C., et al., Molecular characterization of dendritic cell-derived exosomes. Selective accumulation of the heat shock protein hsc73. J Cell Biol, 1999. 147(3): p. 599-610.
  3. Fordjour, F.K., et al., A shared, stochastic pathway mediates exosome protein budding along plasma and endosome membranes. J Biol Chem, 2022. 298(10): p. 102394.
  4. Zhang, M., et al., Proteomics Analysis of Exosomes From Patients With Active Tuberculosis Reveals Infection Profiles and Potential Biomarkers. Front Microbiol, 2021. 12: p. 800807.
  5. Ai, Y., et al., Endocytosis blocks the vesicular secretion of exosome marker proteins. Sci Adv, 2024. 10(19): p. eadi9156.
  6. Luo, A., et al., Exosome-derived miR-339-5p mediates radiosensitivity by targeting Cdc25A in locally advanced esophageal squamous cell carcinoma. Oncogene, 2019. 38(25): p. 4990-5006.
  7. Chen, Y., et al., Extracellular Vesicles Derived from Selenium-Deficient MAC-T Cells Aggravated Inflammation and Apoptosis by Triggering the Endoplasmic Reticulum (ER) Stress/PI3K-AKT-mTOR Pathway in Bovine Mammary Epithelial Cells. Antioxidants (Basel), 2023. 12(12).
  8. Xhuti, D., et al., Circulating exosome-like vesicle and skeletal muscle microRNAs are altered with age and resistance training. J Physiol, 2023. 601(22): p. 5051-5073.
  9. Wang, D., et al., Exosome-encapsulated miRNAs contribute to CXCL12/CXCR4-induced liver metastasis of colorectal cancer by enhancing M2 polarization of macrophages. Cancer Letters, 2020. 474: p. 36-52.
  10. Zhao, S., et al., Tumor-derived exosomal miR-934 induces macrophage M2 polarization to promote liver metastasis of colorectal cancer. Journal of Hematology & Oncology, 2020. 13(1): p. 156.
  11. Zhao, Z., et al., Tumor-derived miR-20b-5p promotes lymphatic metastasis of esophageal squamous cell carcinoma by remodeling the tumor microenvironment. Signal Transduct Target Ther, 2023. 8(1): p. 29.
  12. Tamgue, O., et al., Differential Targeting of c-Maf, Bach-1, and Elmo-1 by microRNA-143 and microRNA-365 Promotes the Intracellular Growth of Mycobacterium tuberculosis in Alternatively IL-4/IL-13 Activated Macrophages. Front Immunol, 2019. 10: p. 421.
  13. Ulger, M., et al., Possible relation between expression of circulating microRNA and plasma cytokine levels in cases of pulmonary tuberculosis. J Infect Dev Ctries, 2022. 16(7): p. 1166-1173.
  14. Andonian, B.J., et al., Plasma MicroRNAs in Established Rheumatoid Arthritis Relate to Adiposity and Altered Plasma and Skeletal Muscle Cytokine and Metabolic Profiles. Front Immunol, 2019. 10: p. 1475.
  15. Ghafouri-Fard, S., et al., The eminent roles of ncRNAs in the pathogenesis of psoriasis. Noncoding RNA Res, 2020. 5(3): p. 99-108.
  16. Valenzuela-Miranda, D., et al., MicroRNA-based transcriptomic responses of Atlantic salmon during infection by the intracellular bacterium Piscirickettsia salmonis. Dev Comp Immunol, 2017. 77: p. 287-296.

Reviewer 2 Report

Comments and Suggestions for Authors

The study focuses on the role of tumor-derived exosomal miR-143-3p in inducing M2 macrophage polarization, which contributes to radiation resistance in advanced esophageal squamous cell carcinoma (ESCC). The innovation of this paper is the development of a predictive model for liquid biopsies, offering a less invasive method and providing a way to obtain tumor-related data from bodily fluids, useful for guiding targeted therapy strategies in patients difficult to biopsy. Researchers found that during radiotherapy, elevated levels of serum miR-143-3p and miR-223-3p indicate radiation resistance and can predict disease progression. Based on these markers, a predictive model was developed, potentially providing personalized treatment for ESCC patients. The study shows that exosomal miR-143-3p produced by ESCC can be internalized by macrophages, leading to M2 polarization, which may promote tumor progression and recurrence. This suggests that miR-143-3p can serve as a non-invasive biomarker for predicting radiotherapy response and assessing patient prognosis.

Major: The study's conclusions are based on a relatively small patient group, which may limit the generalizability of the findings. The study primarily focuses on immediate radiotherapy response and short-term outcomes. Longer follow-up times are needed to assess the impact of these biomarkers on long-term survival rates and quality of life. There is also an expectation for deeper mechanistic insights, not only regarding how ESCC cell-derived exosomal miR-143-3p induces M2 polarization in macrophages but also its effects on protein expression and pathways involved. The current findings are promising, but protein-level breakthroughs would enhance the results.

 Minor: In Figures 1-4, all parts should be labeled as A, B, C, D, etc., without using labels like A1, A2, A3.

Comments on the Quality of English Language

Minor editing of English language required

Author Response

Dear Editor and reviewers,

We sincerely appreciate the careful reviews of our work and the reviewers’ thoughtful comments. We have responded to the reviewers’ comments and have updated the manuscript accordingly. We highlighted our answers in blue for easily reading in Response letter.

However, considering that there is only a 10-day time for revisions, there hasn't been sufficient time to conduct additional experiments. We have made our best effort to respond to the reviewers' comments. If further modifications and improvements are necessary, please do not hesitate to contact us with any further questions.

Furthermore, this manuscript was copyedited by a professional English editing service (Elixigen Co.) that specializes in scientific papers before submission. We have conducted a thorough review and further revised the language. The revised manuscript has been sent to Elixigen Co. for proofreading.

We continue to appreciate your consideration of our manuscript for publication in your esteemed journal.

Sincerely,

Mei Liu, PhD

Laboratory of Cell and Molecular Biology

State Key Laboratory of Molecular Oncology

National Cancer Center/National Clinical Research Center for Cancer/Cancer Hospital

Chinese Academy of Medical Sciences & Peking Union Medical College

Beijing 100021

P.R. China

Comments and Suggestions for Authors

The study focuses on the role of tumor-derived exosomal miR-143-3p in inducing M2 macrophage polarization, which contributes to radiation resistance in advanced esophageal squamous cell carcinoma (ESCC). The innovation of this paper is the development of a predictive model for liquid biopsies, offering a less invasive method and providing a way to obtain tumor-related data from bodily fluids, useful for guiding targeted therapy strategies in patients difficult to biopsy. Researchers found that during radiotherapy, elevated levels of serum miR-143-3p and miR-223-3p indicate radiation resistance and can predict disease progression. Based on these markers, a predictive model was developed, potentially providing personalized treatment for ESCC patients. The study shows that exosomal miR-143-3p produced by ESCC can be internalized by macrophages, leading to M2 polarization, which may promote tumor progression and recurrence. This suggests that miR-143-3p can serve as a non-invasive biomarker for predicting radiotherapy response and assessing patient prognosis.

Major: The study's conclusions are based on a relatively small patient group, which may limit the generalizability of the findings. The study primarily focuses on immediate radiotherapy response and short-term outcomes. Longer follow-up times are needed to assess the impact of these biomarkers on long-term survival rates and quality of life. There is also an expectation for deeper mechanistic insights, not only regarding how ESCC cell-derived exosomal miR-143-3p induces M2 polarization in macrophages but also its effects on protein expression and pathways involved. The current findings are promising, but protein-level breakthroughs would enhance the results.

Response: We totally agree with the reviewer’s comment that the relatively small sample size in our study limited the generalizability of the findings. ESCC is well-known for its aggressive nature. In our cohort, with a median follow-up time of 20 (6–74) months, 67.5% (27/40) of patients experienced disease progression, and 47.4% (19/40) succumbed to ESCC. Though with a relatively limited follow-up, the clear classification of patients into progression and non-progression groups was facilitated by the observation of tumor progression in over half of the patients. Additionally, our research findings indicated that the fold-change of miR-143-3p and miR-223-3p served as a reliable predictor of overall survival in patients with ESCC (Figure 1L, 1N, Table 1).

Furthermore, our research presented a minimally invasive approach for predicting the likelihood of progression or metastasis in patients with ESCC after radiotherapy. It is important to note that our exploration of the underlying mechanisms was not comprehensive. Moving forward, it is essential to conduct further investigations to enhance our comprehension and enhance clinical results for individuals with esophageal squamous cell carcinoma.

We added a sentence in the text to address these limitations (Lines 494-499):

This study had some limitations, including its single-center design, a relatively small sample size, and the lack of a validation dataset. A study of a multi-center large sample size (e.g., n > 100) with longer follow-ups is needed to strengthen generalizability of the findings. Furthermore, a more thorough exploration of the mechanisms by which ESCC cell-derived exosomal miR-143-3p induces M2 polarization in macrophages, along with its subsequent effects on protein expression and associated pathways, is necessary.

 Minor: In Figures 1-4, all parts should be labeled as A, B, C, D, etc., without using labels like A1, A2, A3.

Response: Thank you for your recommendations. The labels have been modified in accordance with the suggestions provided.

Round 2

Reviewer 1 Report

Comments and Suggestions for Authors

The Author responses are appreciated. The responses addressed and help clarify most of the comments. However, there is still an issue with referring to your EV samples as solely exosomes. While it may be true that the "primary constituents" of the EV populations were exosomes, or they were 'mainly exosomes', that does not change the fact that other EVs were included in those samples. Thus, these samples should be referred to as highly-enriched in exosomes, as it's clear they are not a pure, homogeneous population. This may seem like a "small" thing, but it is crucial in scientific literature to be as accurate as possible in descriptions and discussions.

Author Response

Comments and Suggestions for Authors

The Author responses are appreciated. The responses addressed and help clarify most of the comments. However, there is still an issue with referring to your EV samples as solely exosomes. While it may be true that the "primary constituents" of the EV populations were exosomes, or they were 'mainly exosomes', that does not change the fact that other EVs were included in those samples. Thus, these samples should be referred to as highly-enriched in exosomes, as it's clear they are not a pure, homogeneous population. This may seem like a "small" thing, but it is crucial in scientific literature to be as accurate as possible in descriptions and discussions.

Response: We sincerely appreciate the reviewer’s thoughtful comments. In order to ensure the coherence and comprehensiveness of the article and title, we have incorporated the mentioned issue into the "limitations" section of the discussion. The limitations have been included in the revised manuscript as follows (Lines 498-500):

In addition, the primary constituents isolated through the methodologies outlined in our study are referred to as highly-enriched in exosomes,because there is a possibility of other EVs contamination.

Reviewer 2 Report

Comments and Suggestions for Authors

This article can be published.

Author Response

Thank you for your patient help and guidance.